# Effects of a Physical Activity Program Potentiated with ICTs on the Formation and Dissolution of Friendship Networks of Children in a Middle-Income Country

**DOI:** 10.3390/ijerph17165796

**Published:** 2020-08-11

**Authors:** Ana M. Guerra, Felipe Montes, Andrés F. Useche, Ana María Jaramillo, Silvia A. González, Jose D. Meisel, Catalina Obando, Valentina Cardozo, Ruth F. Hunter, Olga L. Sarmiento

**Affiliations:** 1Department of Industrial Engineering, Universidad de Los Andes, Bogotá 111711, Colombia; fel-mont@uniandes.edu.co (F.M.); af.useche10@uniandes.edu.co (A.F.U.); am.jaramillo37@uniandes.edu.co (A.M.J.); v.cardozo10@uniandes.edu.co (V.C.); 2Social and Health Complexity Center, Universidad de Los Andes, Bogotá 111711, Colombia; jose.meisel@unibague.edu.co; 3Department of Computer Science, University of Exeter, Exeter EX4 4PY, UK; 4Healthy Active Living and Obesity Research Group, Children’s Hospital of Eastern Ontario Research Institute, Ottawa, ON K1H 8L1, Canada; sa.gonzalez68@uniandes.edu.co; 5School of Medicine, Universidad de los Andes, Bogotá 111711, Colombia; osarmien@uniandes.edu.co; 6Facultad de Ingeniería, Universidad de Ibagué, Carrera 22 Calle 67, Ibagué 730001, Colombia; 7INRIA, Institut du Cerveau et de la Moelle épinière, ICM, Inserm U 1127, CNRS UMR 7225, Sorbonne Université, 75013 Paris, France; cobando85@gmail.com; 8Centre for Public Health, School of Medicine, Dentistry and Biomedical Sciences, Queen’s University Belfast, Belfast BT7 1NN, UK; ruth.hunter@qub.ac.uk

**Keywords:** physical activity, technology, QAP logistic regression, STERGM, friendship ties, social network analysis, eHealth, mobile phone interventions

## Abstract

This paper assesses the potential cohesion effect of a physical activity (PA) school-based intervention potentiated using text messages (SMS) through analyzing longitudinally the friendship network structure and the mechanisms of the formation and dissolution of friendships. Three schools (*n* = 125 participants) in Bogotá, Colombia, were randomly assigned into three groups: Modulo Activo Recreo Activo (MARA) + SMS (networks 1 and 2), MARA (networks 3 and 4), and control (no intervention: networks 5–7). We collected socio-economic, health-related, network structure, and intervention satisfaction variables in the baseline and after 10 weeks on July–November 2013. For each classroom network, we conducted four models using a temporal and static network approach to assess (1) temporal social network changes, (2) friendship homophily, (3) friendship formation and dissolution mechanisms, and (4) effect of SMS on the networks’ cohesion. We found that (1) social cohesion emerged in the four intervened networks that were measured over time with transitivity and homophily driven by clustering, (2) the intervention affected the mechanisms of friendship formation and dissolution, and (3) MARA + SMS on average created more social cohesion and 3.8 more friendships than the program alone. Potentially, school-based interventions with information and communication technologies (ICT) such as MARA + SMS could encourage social cohesion among children. The particular characteristics of each school network need to be considered when developing school-based interventions.

## 1. Introduction

Childhood is a crucial period for developing healthy lifestyles [1] such as regular engagement in physical activity (PA). Physically active children have better physical, psychological, and cognitive health indicators [2]. However, 80% of children worldwide and 70% of children in Colombia are insufficiently physically active [3,4]. Physical inactivity and sedentary behaviors have been independently associated with obesity, poor cardio-metabolic health, and poor psychosocial health [4]. Longitudinal studies have found that PA levels decrease during adolescence, and thus there is a need to understand the mechanisms of interventions that promote long-term healthy lifestyles [5,6].

Physical education classes and recess time are relevant for the promotion of PA in students since they spend 6 to 8 hours per day in school [7,8]. Consequently, strategies involving recess time are promising interventions for the promotion of PA in students [9,10], and in Latin America, school interventions are recommended as effective strategies [11]. In Bogotá, Colombia, there is evidence about the Modulo Activo Recreo Activo (MARA), a PA intervention using information and communication technologies (ICT). Children attending schools with the MARA program increased the average minutes of PA compared to control (Difference(*T*_1_ − *T*_0_) = 6.1 min, SE = 3.49, *p* = 0.005) [7]. However, there are no studies on the effect of MARA on the formation and dissolution of friendship networks over time, and the potential of the program in fostering social cohesion when potentiated with ICT.

Among school-age children, studies on social networks conducted mainly in high-income countries evidence that children may be significantly influenced by their friends’ PA levels and obesity-related behaviors [12,13,14]. Likewise, evidence of peer influence on healthy behaviors has shown that physically active peers have a positive impact on adolescents’ PA [15,16]. A longitudinal study found that adherence to PA guidelines among children increased by 6% for each additional active friend [17]. Although the importance of social networks and peers influence in healthy behaviors, there is limited evidence on the longitudinal mechanisms that underlie the formation and dissolution of friendships, especially in populations in low- and middle-income countries (LMICs). Previous studies have shown that the use of inherent social networks structures in schools may provide a cost-effective way of encouraging children to be more physically active [9]. However, there are gaps regarding the potential longitudinal synergic effect of interventions and friendships on the cohesion of the social networks and to promote children’s PA.

In this context, the randomized school-based intervention MARA provided a unique opportunity to assess friendship networks structures longitudinally in a program aimed at promoting PA with ICT. This project, in addition to measuring health-related and network-structure variables, also measured relevant socio-economic and satisfaction variables. Socio-economic variables could be related to the structure of the network and according to socio-ecological frameworks could influence the PA levels of children [18,19]. Intervention satisfaction variables are related to motivation and engagement in the program [20] and could also be related to the structure of the network. This study aims to longitudinally assess the network structure, the mechanisms of the formation and dissolution of friendships, and the potential cohesion effect of the intervention MARA. For this, a four-step analysis was carried out. First, we measured temporal social network changes to determine if a temporal analysis of children’s networks is feasible [21]. Then, we analyzed the friendships’ homophily [22] in order to determine what are the attributes that define friendships between children before and after the intervention [23,24]. Next, we evaluated friendship formation and dissolution mechanisms over time [25]. Lastly, we evaluated the effect of text messages (SMS) on network cohesion to determine if the use of ICT generates social benefits in children’s networks [26]. Findings and recommendations from this study could be used as a guideline in future practice or research in school settings.

## 2. Methods

### 2.1. Intervention

We designed a randomized school-based intervention in fifth-grade students from three different schools in Bogotá, Colombia. Schools were randomly selected from the 20 eligible schools participating in the International Study of Childhood Obesity, Lifestyle and Environment (ISCOLE) [27]. The three schools were randomly assigned into two intervention groups (MARA and MARA + SMS) and one control group (Figure 1). For the 10 weeks, from July to November of 2013, the local authority for sports and recreation—District Institute of Recreation and Sports (acronym in Spanish: IDRD) implemented the intervention. The MARA group was composed of two classrooms from one school. This group was intervened during recess time. During the usual school recess of 30 min, we implemented optional PA sessions three times per week. A total of 30 sessions of PA combined with supervised games with ties, balls, hoops, stairs, parachutes, rugs, and dancing were carried out. The MARA + SMS group was composed of two classrooms from another school. In addition to the intervention, this group was targeted with SMS each weekday, promoting the students’ participation, as well as their engagement, motivation and empowerment, enhancing the diffusion of information about extracurricular PA and healthy behaviors among their classmates and family members. The SMSs were designed by psychologists and were sent by cellphone to the students or their parents, being subject to the approval of the parents and the instructions of the *Universidad de Los Andes* Institutional Review Board (IRB). There were four categories of SMS: tip, testimony, motivation, and empowerment messages about healthy behaviors and PA. The control group was composed of three classrooms from the remaining school. This group did not receive any intervention, except for the data collection process. Data were collected at two points in time: a baseline before the intervention (*T*_0_) and during the 10th week of the intervention (*T*_1_). This study was approved by the Institutional Review Board (IRB) of Universidad de Los Andes (Process 214–2013). Parents or guardians of potential participants, potential participants themselves, and schools received informed consent in order to authorize their participation in the study. Additional details of the study design, sample, and methodology have been published elsewhere [7].

### 2.2. Data Collection

We collected socio-economic, health-related, social network structure, and intervention satisfaction variables during July–November, 2013. In the three schools, health-related measures were obtained according to ISCOLE protocols [27]. Data were collected using a questionnaire designed to evaluate the children’s social network structure and (un)healthy behaviors. Additionally, parents completed questionnaires about socio-demographic information.

### 2.3. Measures

#### 2.3.1. Socio-Economic Variables

The socio-economic variables included age, sex, mother’s educational level, and household income. Age was computed from date of birth and the date of anthropometry measurements. Sex and parental education were recorded on the demographic and family health questionnaire. The maternal education variable included the following categories: less than high school, incomplete high school, complete high school or some college, and university degree or postgraduate degree.

#### 2.3.2. Health-Related Variables

Bodyweight was measured using a portable Tanita SC-240 Body Composition Analyzer (Tanita, Arlington Heights, IL, USA), and height was measured with a Seca 213 portable stadiometer (Seca, Hamburg, Germany). Using the 2007 World Health Organization (WHO) growth reference tables, we calculated body mass index (BMI), categorizing it as underweight (BMI Z-score [BMIZ] < −2 SDs), normal weight (−2 SDs ≤ BMIZ ≤ 1 SD), or overweight (BMIZ > 1 SD) [28].

PA levels were measured by the minutes spent performing moderate to vigorous PA (MVPA) using an Actigraph GT3X+ accelerometer (ActiGraph, Pensacola, FL, USA), calculated with the Evenson equation [29]. Participants were asked to wear the accelerometer 24 h/day for 7 days. A valid measurement was defined by a minimum wear time of four valid days with at least 10 valid h/day [27]. We classified the children as meeting the PA recommendation or not meeting recommendations using the WHO threshold of 60 min of MVPA/day [10].

#### 2.3.3. Network Structure Variables

Students nominated their friends from a list of classmates who were also participating in the intervention. Students could only nominate participants in the same classroom, and there was no limit to the number of nominations. The phrasing of the network question was as follows:
We would like to know who are your friends and with whom you like to share your time.
Below you will find a list with the names of your classmates who are participating in the study. First, we want to know which girls are your friends. Look at the left column and in the box next to the name of the girl who is your best friend write a ‘1’. In the box next to the name of the girl who is your second best friend write a ‘2’. In the box next to the name of the girl who is your third best friend write a ‘3’. In the box next to the name of any of the girls that are your friends write ‘4’.
Please, just write a ‘1’, a ‘2’ and a ‘3’. If you don’t want to, you don’t have to put a ‘2’ or a ‘3’. You can write ‘4’ as many times as you want. Just make sure those girls are your friends. When you’re done, do the same with the boys, in the right column.

On the basis of the responses, we built a directed network per classroom; nodes represent children, and directed ties connect two nodes if a child nominated another as a friend. We built seven networks, one per classroom, for each period *T*_0_ and *T*_1_.

For each classroom network, we calculated five different parameters (Table 1): (1) in-degree, (2) out-degree, (3) degree, (4) clustering coefficient, and (5) closeness centrality. All these variables were included in all models, except in the separable temporal exponential random graph model (STERGM).

#### 2.3.4. Intervention Satisfaction Variables

Students reported their participation in the program and their satisfaction level with the intervention. Satisfaction indicators included participation (1 if attended, 0 otherwise), enjoying time (“I have fun, and I meet people at recess thanks to MARA” (enjoying time): 1 if yes, 0 otherwise), PA engagement (“I do PA at recess time thanks to MARA” (do PA): 1 if yes, 0 otherwise), safe play (“I feel safe when I play at recess thanks to MARA” (recreation safety): 1 if yes, 0 otherwise), and healthy experience (“recess is a healthy experience thanks to MARA”: 1 if yes, 0 otherwise). These questions were designed collectively with the IDRD as part of the evaluation of the MARA program, taking into account the relevance of the satisfaction for the success of an intervention [20].

### 2.4. Temporal and Static Network Analysis

For each classroom network, we conducted four different models using a temporal and static network approach to assess (1) temporal social network changes, (2) friendship homophily, (3) friendship formation and dissolution, and (4) effect of SMS on the networks’ cohesion.

#### 2.4.1. Temporal Social Network Changes

We employed temporal network analysis to assess the change in the friendship networks between *T*_0_ and *T*_1_ and investigated the pertinence of analyzing its effect. We calculated the Jaccard index $J_{i}$ that measures the change in the ties structure of each network i over time [21].
(1)$$J_{i}=\frac{E_{\left(1,1\right)}}{E_{\left(1,0\right)}+E_{\left(0,1\right)}+E_{\left(1,1\right)}}$$

$E_{\left(1,1\right)}\;$represents the number of ties in both periods, $E_{\left(0,1\right)}$ is the number of new ties, and $E_{\left(1,0\right)}$ is the number of dissolved ties. A value lower than 0.3 for $J_{i}$ indicates that the dissolution of ties increased more than the formation of ties. A value higher than 0.6 for $J_{i}$ indicates that the formation of ties increased more than the dissolution of ties. Values between 0.3 and 0.6 are acceptable to conclude that there is a meaningful change in friendship ties among periods [21].

#### 2.4.2. Friendship Homophily

Homophily, or assortativity, is defined as people’s “tendency to associate with others whom they perceive as being similar to themselves in some way” [22]. We assessed the existence of homophily at a dyadic level for each time [22]. To determine if friendship nominations were associated with the individual’s characteristics, we conducted a quadratic assignment procedure (QAP) logistic regression that estimated the probability $P\left(Y_{ij}=1|X\right)$ of existence of a tie between two nodes (i,j) explained by a specific nodal attribute X [23,24]. Quadratic assignment procedure (QAP) logistic regression was used instead of traditional ordinary least squares (OLS) logistic regression, because the OLS method assumes dyadic independence, and in social networks, dyads are interdependent [23]. QAP randomly rearranges the rows and their matching columns of one of the data matrices. The resulting matrix is independent of the original and has the same properties [24]. Regarding the variables, the mother’s education level was categorized to better understand the effect of this variable in the logistic regression as a continuous variable.
(2)$$P\left(Y_{ij}=1|X\right)=\frac{1}{1+e^{-\beta_{0}+C_{d}*\sum_{d\;\in \;D}\beta_{d}*\delta \left(x_{i}^{d},x_{j}^{d}\right)+\left(1-C_{d}\right)*\sum_{d\;\in \;D}\beta_{d}*|x_{i}^{d}-x_{j}^{d}|}}$$

$Y_{ij}$ is 1 if there is a tie between nodes i and j, otherwise it is 0. D is the collection of all attributes indexed with d. $C_{d}$ takes the value of 1 if the variable *d* is categorical, or 0 otherwise. $\delta \left(x_{i}^{d},x_{j}^{d}\right)$ denotes the Kronecker delta, which takes the value of 1 if $x_{i}$ and $x_{j}$ have the same value for the attribute d, or 0 otherwise. $\left|x_{i}^{d}-x_{j}^{d}\right|$ is the absolute difference between $x_{i}$ and $x_{j}$. We estimated the regression coefficients $\beta_{d}$ by maximum likelihood estimation. Exponentiated coefficients can be interpreted as odds ratios such that a unit difference in predictor $x^{d}$ corresponds to a multiplicative change of $e^{\beta_{d}}$ in the odds.

#### 2.4.3. Friendships’ Formation and Dissolution

We assessed the friendship network dynamic processes of formation and dissolution of ties attributed to the interventions. We implemented a separable temporal exponential random graph model (STERGM), which separates the events of connection and disconnection of nodes in the discrete time periods $T_{0}$ and $T_{1}$ [25]. This model assumes that the formation and dissolution of friendship ties are independent events, which means both processes can occur for different reasons; therefore, they must be analyzed separately [25]. For each time t, this procedure considers the formation formula $P(Y^{+}=y^{+}|\;Y^{t-1}=y^{t-1};\theta )$ and the dissolution formula $P(Y^{-}=y^{-}|\;Y^{t-1}=y^{t-1};\theta )$. This model assumes that the formation and dissolution of friendship ties are independent events, which means that the formation and dissolution of friendships can occur for different reasons and therefore must be analyzed separately [25].
(3)$$P(Y^{+}=y^{+}|\;Y^{t-1}=y^{t-1};\theta )=\frac{\exp\left(\eta^{+}\left(\theta^{+}\right)\cdot g^{+}\left(y^{+},y^{t-1}\right)\right)}{C_{\eta^{+}},g^{+}\left(\theta^{+},y^{t-1}\right)},\;y^{+}\epsilon \;Y^{+}\left(y^{t-1}\right)$$
(4)$$P(Y^{-}=y^{-}|\;Y^{t-1}=y^{t-1};\theta )=\frac{\exp\left(\eta^{-}\left(\theta^{-}\right)\cdot g^{-}\left(y^{-},y^{t-1}\right)\right)}{C_{\eta^{-}},g^{-}\left(\theta^{-},y^{t-1}\right)},\;y^{-}\epsilon \;Y^{-}\left(y^{t-1}\right)$$

$Y^{+}$ is the formation network, and the STERGM is controlled by a *p*-vector of formation parameters ($\eta^{+}\left(\theta^{+}\right))$, normalizing constants $\left(C_{\eta^{+}},g^{+}\left(\theta^{+},y^{t-1}\right)\right)$, and statistics ($g^{+}\left(y^{+},y^{t-1}\right)$) (Equation (3)). $Y^{-}$ is the dissolution network controlled by a *q*-vector of dissolution parameters $\left(\eta^{-}\left(\theta^{-}\right)\right)$, normalizing constants $\left(C_{\eta^{-}},g^{-}\left(\theta^{-},y^{t-1}\right)\right)$, and statistics ($g^{-}\left(y^{-},y^{t-1}\right)$) (Equation (4)) [25].

After fitting the model over real data, the resulting parameters measure the effect on the incidence or duration of ties in terms of the log-odds metrics. In the formation model, a positive parameter $\theta^{+}$ means an increase in the probability of forming ties between nodes with similar features; a negative $\theta^{+}$ indicates an increase in the probability of forming less ties between nodes with similar features. In the dissolution model, a positive parameter $\theta^{-}$ means an increase in the probability of preserving ties between nodes with similar features, and a negative $\theta^{-}$ indicates an increase in the probability of dissolving ties between nodes with similar features [25].

The models were fitted using the Markov Chain Monte Carlo and the conditional maximum likelihood estimator [25]. These models included network structural variables and individual attributes of the nodes (Table 2). To run this model, we used the stergm package [30]. To assess the level of significance of the variables, we used a confidence level of 90%. Seven STERGMS were built, one for each network. We found that BMI should not be included in the model when studying control networks because this generated poor convergence in the model. This means that the model parameters were not able to explain the mechanisms of formation and dissolution of friendships in control networks, and therefore the model did not fit over real data. With this change, all models achieved good convergence. That is, all the models fit over real data and were able to explain the mechanisms of formation and dissolution of friendships in the networks.

#### 2.4.4. Effect of SMS on the Network Cohesion

We performed a difference in difference (DD) model to the effect of SMS over MARA on the networks’ cohesion by comparing the results of *T*_0_ with *T*_1_ [26] for the closeness, degree (total, in, and out), and clustering coefficient, which are network parameters that can be used to measure cohesion. We conducted a DD with (1) MARA schools as the intervention group and control schools as the control, (2) MARA + SMS schools as the intervention group and control schools as the control group, and (3) MARA + SMS schools as the intervention group and MARA schools as the control group. The DD method assumes that the populations compared are statistically equal and estimates the average intervention impact as follows [26]:(5)$$y_{i}=\beta_{0}+\beta_{1}*t_{i}+\beta_{2}*T_{i}+\beta_{3}*t_{i}*T_{i}$$
where $y_{i}$ are the values of the network parameters for the node i. $t_{i}$ takes the value of 1 if the node i received the intervention and 0 otherwise. $T_{i}$ takes the value of 1 if the results are for $T_{1}$ time and 0 if the results are for $T_{0}$ time. $\beta_{3}$ represents the impact of the intervention on the result variable among time [26].

## 3. Results

The results of the models are summarized in Appendix A. The sample included 71 girls and 54 boys. The children’s age ranged from 9 to 13 years. The body mass index (BMI) ranged from 13.3 to 27.75 kg/m^2^. The PA levels varied from 19.92 to 143.9 of moderate to vigorous MVPA minutes/day (Table A1). Although there was variability, there were no significant differences in the baseline [7].

### 3.1. General Results

#### 3.1.1. Temporal Social Network Changes

The Jaccard coefficient ranged from 0.31 to 0.4 for the different networks, showing that all the networks presented an appropriate level of change in friendship ties among time periods, meaning that it was appropriate to conduct longitudinal network analysis [21] (Table A2). Additionally, in 4/4 of the intervened networks, the average clustering coefficient increased (from 0.27 to 0.28 in network 1, from 0.29 to 0.32 in network 2, from 0.28 to 0.29 in network 3, and from 0.28 to 0.38 in network 4) whereas the control networks did not show this general pattern (Table A2).

#### 3.1.2. Friendship Homophily

##### Socio-Economic Variables

We found significant dyadic homophily driven by sex for most networks, meaning that children were more likely to relate with same-sex peers than expected by chance. In *T*_0_, this effect was significant in at least one of the networks in each school (networks: 1, 2, 4, 6, 7), and, in T_1_, the effect was still significant in these networks (Table A3). We did not find more overall significant effects regarding socio-economic variables on the friendship homophily among children.

##### Health-Related Variables

We did not find overall significant effects of homophily attributable to health-related variables. This means that there are no health-related variables that explain in general the friendship homophily among children.

##### Network Structure Variables

We found significant assortativity effects driven by the clustering coefficient in 1/2 MARA + SMS networks and 2/2 MARA networks (Table A3). In *T*_0_, this effect was significant in 1/2 MARA networks, and in *T*_1_ the effect was significant in 1/2 MARA networks and 2/2 MARA + SMS networks (Table A3). This means that children relate with those who have similar levels of cohesion in their ego networks for the intervened networks.

We did not find significant overall assortativity effects driven by network structure variables to analyze in the control group. This implies that the network structure has no effect on friendship homophily among children in the control group.

#### 3.1.3. Friendship Formation and Dissolution

##### Network Structure Variables

Transitivity was significant and positive in the formation of friendship ties in the intervened networks (MARA + SMS and MARA) and in only one-third of control networks (Table A4). Additionally, transitivity was significant and positive in the dissolution of friendship ties in two-quarters of the intervened networks (one-half MARA networks and one-half MARA + SMS networks) and in two-thirds of control networks (Table A4). The latter results mean that children relate with those who have similar levels of cohesion in their ego networks in all intervened networks and just one of the control networks.

##### Socio-Economic, Health-Related, and Satisfaction Variables

We did not find overall formation and dissolution effects regarding socio-economic, health-related, and intervention satisfaction variables in intervened and control networks (Table A4). This means that socio-economic, health-related, and satisfaction variables do not present an overall effect that allows us to conclude that they generally affect the mechanisms of formation and dissolution of friends.

#### 3.1.4. Effect of SMS on the Network Cohesion

The results of the DD model showed that the popularity measures (in-degree, out-degree, degree) were significant and positive when adding technology to the intervention (MARA + SMS vs. MARA). This result suggests that SMS has an impact on the creation of more friendship ties for the intervention. The closeness estimate was positive and significant in MARA + SMS, suggesting that the effect of the intervention has an impact in reducing the number of friendships required for a child to connect with others. We did not find overall significant effects when comparing each intervention with the control (Table A5). This means that receiving the intervention does not have a significant effect on creating friendships and increasing cohesion in the networks, but the fact of using SMS does have a positive effect.

### 3.2. Classroom Network-Specific Results

Each classroom network has its particularities according to the context, and thus we found specific results for each one (Figure 2 and Table 3 and Table 4). First, there were significant changes in connectivity between both periods on the basis of on the intervened school. Second, all the networks had specific structural changes. Third, we found significant dyadic assortativity driven by age in two out of the two MARA + SMS networks in T_1_, and dyadic assortativity driven by PA in one-half of MARA + SMS networks (networks 1,2) in *T*_1_, and dyadic assortativity driven by the fact that MARA is a program that allows children to do PA at recess time in one-half of the MARA networks (network 3). Lastly, we found that the absolute difference in PA was significant and negative in the formation of friendship ties in two-quarters of the intervened networks (networks 2 and 3), BMI was significant and negative in the dissolution of friendship ties in one-half of the MARA + SMS networks (network 1), and the fact that children have fun and meet people at recess (enjoying time) and do PA were significant and positive in the formation of friendship ties in two-quarters of the intervened networks (network 4 for enjoying time and network 1 for doing PA).

## 4. Discussion

This study, for the first time, evaluated longitudinally the network structure, the mechanisms of the formation and dissolution of friendships, and the potential cohesion effect of a school-based governmental PA program in Latin America. The study shows that MARA, a school-based PA intervention that uses ICT, could promote social co-benefits that consist in increasing cohesion in the friendship network and encouraging children to relate with those who have similar levels of cohesion in their individual networks. The study also underscores the heterogeneity among school networks in friendship formation and cohesion that could be explained in part by the acceptability and adherence of the program. Our results provide evidence of the co-benefits of school-based interventions intended to promote healthy behaviors among children living in vulnerable settings.

We found that MARA + SMS could increase cohesion in the friendship network and encourage children to relate with those who have similar levels of cohesion in their individual networks. The potential impact of this benefit is explained in previous studies in high-income countries that have shown that cohesive children networks are related to lower levels of BMI [31], and cohesive classrooms promoting stable relationships generate sustainable behavior patterns among children [32]. Therefore, school-based interventions such as MARA could increase the level of cohesion and connectivity between children while facilitating stable environments and the dissemination and sustainable adoption of healthy behaviors.

When comparing the group that received SMS (MARA + SMS) with the program without SMS (MARA), we found that when ICT is included, the impact on increasing network cohesion is greater. SMS had an impact on the creation of friendship ties and in increasing closeness between children. Our results show that the inclusion of SMS in the intervention caused, on average, 3.8 additional friendships to form, and that there was 0.01 closeness between the children of the MARA + SMS networks. In a network where there is more closeness, the information can flow more quickly. These results are consistent with the literature, as previous studies have shown the potential in using ICT to promote healthy lifestyles and health-related structural changes that encourage healthy behaviors through immediate feedback and advice [33]. These studies showed that when using ICT to influence children’s behavior, positive effects can be obtained for promoting PA behavior change [33]. Some interventions, mostly from high income countries, showed that the use of technology significantly increases interactions and produces collaborative patterns among children [34]. Our study expands in the impact of the ICT interventions in promoting healthy behaviors in low-income communities in a LMIC, as we found that ICT increases the cohesion of the network.

Moreover, unlike the findings on high income countries [14,35,36,37], in the studied networks, we did not find an overall effect of assortativity driven by PA before and after MARA. This means that, overall, friendships between children were not formed because they shared similar levels of PA. This finding could be explained in part by the fact that PA occurred in our studied schools mostly through play and dancing, and the fact that no competitive sports were evaluated. These low–middle-income schools do not have specific sport clubs that allow children to form groups according to specific sports. Furthermore, we did not find an overall effect with socio-economic variables besides sex and BMI, which could be due to the fact that most of the children were low to low–middle socioeconomic status (SES) and 65% of children had normal BMI.

When analyzing each school in particular, we found heterogeneity in the assortativity effects and in the mechanisms of formation and dissolution of friendships. Specifically, the program had an effect on friendship homophily and in the mechanisms of formation and dissolution of friendships in the networks. We found that PA levels and BMI began to have effects on friendship homophily and in the formation and dissolution of friendships in most of the intervened networks among time. Children became more likely to stop being friends with children with different BMIs (network 1 of MARA + SMS) and became less likely to become friends with children with different PA levels (network 2 of MARA + SMS and network 3 of MARA). This may be due to the fact that the children could have started to relate with others that had an affinity in PA or BMI. Although the purpose of MARA is to promote PA in an inclusive way, there may be a risk that less-active children will begin to group with sedentary instead of grouping with active children if not all of the children participate in the program. In fact, a previous study showed that there may be heterogeneity in the effects of PA on the network structure and on the processes of formation and dissolution of friendships [9]. This means that the processes of selecting friends can change, and therefore a static PA intervention could be inefficient for promoting PA [9]. When designing interventions, it is necessary to take into account the characteristics of the places where the intervention will take place because the network structure can be heterogeneous, and is important to avoid going contrary to the dynamics of the relationships [9]. Given this, we observed that in the schools that we studied that the context and the dynamics of relationships are different than high-income countries, and each course has its particularities and heterogeneity, and thus it is relevant to consider the friendship drivers when designing and evaluating a program such as MARA. Currently, MARA is implemented in 72 schools. The current study provides additional co-benefits that could help the IDRD for advocating the program.

There were some limitations in this study. First, regarding the sample, the size was reduced due to the exclusion of participants that did not provide valid accelerometer data (Figure 1). It is possible that children with an important role within the social network were not considered in the study. However, there was a total response rate of 91% in the baseline, and at the end of the study, the total response rate was 75%, and thus a significant percentage of the population was considered. Additionally, in the MARA + SMS intervention, 68% of the children did not have mobile phones and did not directly receive the SMS, but 98% of them reported to receive the messages through their parents. Second, in the DD model, there were variables that did not present statistical homogeneity. To face this limitation, we controlled for these variables in the DD model. Third, at first, the control group STERGM models did not converge, so we had to exclude BMI to obtain a good level of convergence in the models, and we used a diversity of models that allowed us to analyze cohesion from multiple perspectives. It is important to notice that the robustness of the STERGM models depends on a wide variety of factors that are difficult to isolate, such as the type of variables used and the size of the network [38]. Finally, the study was conducted in 2013 and ICT technology has varied since then. In Colombia, SMS continues to be very relevant, especially for low- to middle-income populations. According to the Ministry of Information Technology and Communications, during the first quarter of 2013, 4 million SMSs were sent [39], while for the fourth quarter of 2019, 457.2 million SMSs were sent [40]. This represents a decrease in the use of SMS by Colombians but also shows that the use continues. Regarding this, it is important to keep in mind that by 2019, 61.3% of Colombians had access to mobile internet and there were 131.6 mobile phone lines for every 100 habitants [40]. This means that a significant percentage of the population still does not have access to mobile internet but does have access to mobile telephony, which is why an intervention that uses SMS as a means of communication is relevant. This may be because Colombia is an LMIC, and in vulnerable environments, SMS could still be a relevant tool.

## 5. Conclusions

School-based programs aimed to promote PA such as MARA could modify health-related behaviors among children, especially as a mechanism to change network structure. The use of ICT for potentiating the intervention could promote the creation of new friendships between children and encourage the cohesion of the network, therein closing the information gap between children. By closing the information gap, it may be easier to spread healthy habits through networks and influence children’s behaviors as a result. The particular characteristics of each school network need to be considered when developing school-based interventions.

## Figures and Tables

**Figure 1 ijerph-17-05796-f001:**
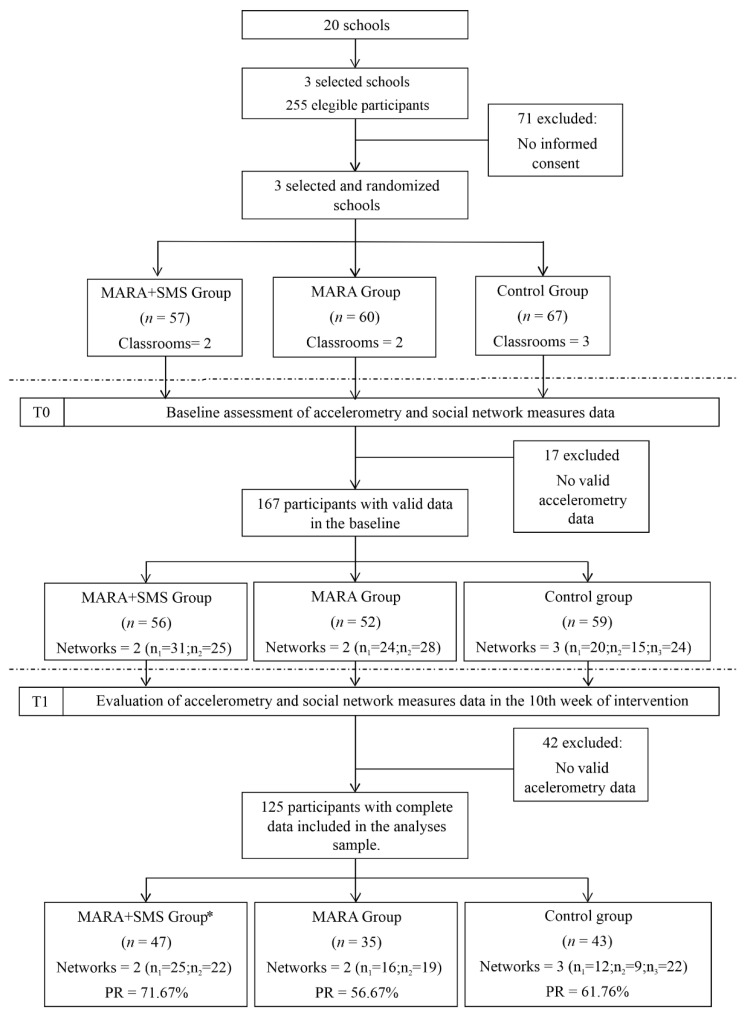
Consort flow diagram of the intervention. Modulo Activo Recreo Activo (MARA) is the Spanish acronym for “active and recreo-active module”, and PR means participation ratio. * A participant dropped out before week 10, but he/she was included in the analysis with the intention-to-treat analysis.

**Figure 2 ijerph-17-05796-f002:**
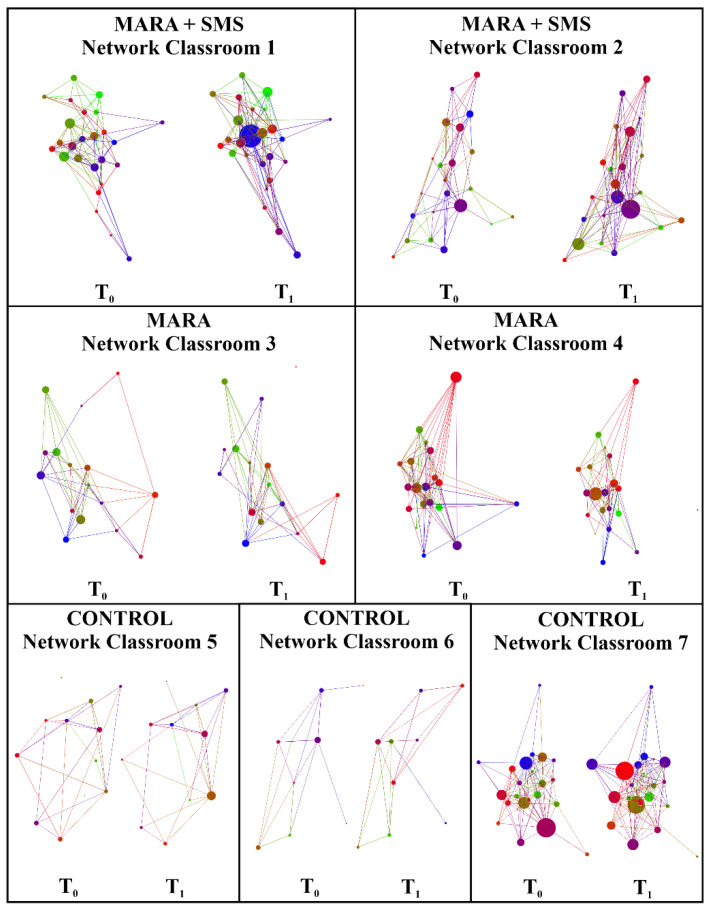
Friendship networks before (*T*_0_) and after (*T*_1_) the intervention of the MARA + SMS, MARA, and control schools. For nodes: color is the child, and size is the in-degree of the child. For edges: color is the color of the targeted child.

**Table 1 ijerph-17-05796-t001:** Network structure variables and their description.

	Parameter	Description	Tie Configuration
1	In-degree	Number of times that the student was nominated by others.	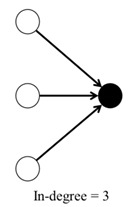
2	Out-degree	Number of times that the student nominated others.	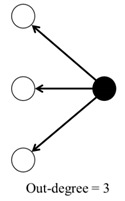
3	Degree	Sum of the in-degree and out-degree.	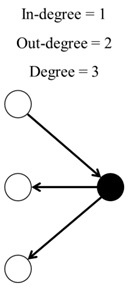
4	Clustering coefficient	Measures the transitivity as the probability that two students sharing a friendship will form a complete triangle with other students, indicating the likelihood that friends of friends will become friends. For each network, the average clustering coefficient ranges from 0 to 1, where 1 indicates that all of the friends of children are also friends with each other.	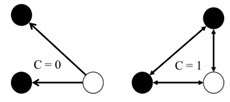
5	Closeness centrality	Mean shortest distance from a student to all others.	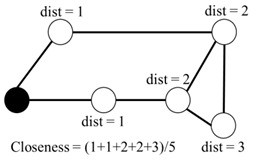

**Table 2 ijerph-17-05796-t002:** Parameters used in the separable temporal exponential random graph models (STERGMs) to measure the formation and dissolution of friendship ties.

	Parameter	Description
1	Geometrical weighted in degree (gwidegree)	Measures the dispersion of edges on the basis of the popularity of each node, following a non-preferential attachment.
2	Geometrical weighted edgewise shared partner (gwesp)	Measures the probability that the network changes due to the transitivity when two connected nodes increase their number of shared partners by 1.
3	Absolute difference physical activity (Absdiff Physical Activity)	Measures the absolute difference of physical activity between a pair of nodes.
4	Absolute difference nutritional state (Absdiff Nutritional State)	Measures the absolute difference of nutritional state between a pair of nodes.
5	Absolute difference BMI (Absdiff BMI)	Measures the absolute difference of BMI between a pair of nodes.
6	Absolute difference participation (Absdiff Asist)	Measures the absolute difference of participation between a pair of nodes.
7	Absolute difference healthy experience (Absdiff V10M_3_4)	Measures the absolute difference of healthy experience between a pair of nodes.
8	Absolute difference recreation safety (Absdiff V10M_3_3)	Measures the absolute difference of recreation safety between a pair of nodes.
9	Absolute difference doing PA (Absdiff V10M_3_2)	Measures the absolute difference of the variable do PA between a pair of nodes.
10	Absolute difference enjoying time (Absdiff V10M_3_1)	Measures the absolute difference of enjoying time between a pair of nodes.

Note: BMI denotes Body Mass Index and PA denotes Physical Activity.

**Table 3 ijerph-17-05796-t003:** Summary of the specific results for each friendship network.

Group	Network Classroom	Temporal Social Network Changes (Table A2 and Figure 2)	Friendship Homophily (Table A3)	Friendship Formation and Dissolution (Table A4)	Effect of SMS on the Networks’ Cohesion (Table A5)	Context Characteristics and Acceptability of the Program
**MARA + SMS**	1	The average degree increased from 9.16 to 11.8.	Children were more likely to connect with same-age peers than expected by chance after MARA (*p*-value < 0.1).	(1) Children became more likely to become friends with children that like the program because they do PA at recess time (*p*-value < 0.05; *p*-value = 0.02). (2) Children became more likely to stop being friends with children with different BMI (*p*-value < 0.05; *p*-value = 0.02).	No specific significant effects were found (*p*-value > 0.1).	According to a qualitative assessment of the program through semi-structured interviews, children from the SMS strategy reported that receiving the SMS reinforced self-esteem and reported connecting with children of their same age during recess time, during MARA, and during non-MARA activities [7].
	2	The average degree increased from 8.18 to 11.	Children were more likely to connect with same-PA peers than expected by chance after MARA (*p*-value < 0.05).	Children became less likely to become friends with children with different PA (*p*-value < 0.1; *p*-value = 0.08).	No specific significant effects were found (*p*-value > 0.1).	
**MARA**	3	The average degree decreased from 7.87 to 7.50.	After MARA, children were more likely to connect with same age peers (*p*-value < 0.05) and peers that also thought that is a program that allows them to do PA at recess time.	Children became less likely to become friends with children that have a different PA (*p*-value < 0.1; *p*-value = 0.08).	No specific significant effects were found (*p*-value > 0.1).	This school had higher acceptability and higher participation (34.4% vs. 12.1%) in the MARA program [7].
	4	The average degree decreased from 11.26 to 9.05.	No specific significant effects were found (*p*-value > 0.1).	Children became more likely to become friends with other children that think they are having fun and meeting people at recess time thanks to the program (*p*-value < 0.05; *p*-value = 0.04).	No specific significant effects were found (*p*-value > 0.1).	

**Table 4 ijerph-17-05796-t004:** Specific results for each control network.

Group	Network Classroom	Temporal Social Network Changes	Friendship Homophily	Friendship Formation and Dissolution	Effect of SMS on the Networks’ Cohesion	Context Characteristics and Acceptability of the Program
**CONTROL**	5	The average degree increased from 5.5 to 5.6 and the average clustering coefficient increased from 0.22 to 0.37.	No specific significant effects were found (*p*-value > 0.1).	No specific significant effects were found (*p*-value > 0.1).	No specific significant effects were found (*p*-value > 0.1).	N/A
	6	The average degree increased from 4.44 to 5.77.	No specific significant effects were found (*p*-value > 0.1).	No specific significant effects were found (*p*-value > 0.1).	No specific significant effects were found (*p*-value > 0.1).	
	7	The average degree increased from 11.8 to 13.54 and the average clustering coefficient decreased from 0.39 to 0.37.	No specific significant effects were found (*p*-value > 0.1).	No specific significant effects were found (*p*-value > 0.1).	No specific significant effects were found (*p*-value > 0.1).	

Note: N/A denotes not applicable.

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
