# Peer review of "Effects of a Physical Activity Program Potentiated with ICTs on the Formation and Dissolution of Friendship Networks of Children in a Middle-Income Country"

_ijerph, 2020, doi:10.3390/ijerph17165796_

Round 1

Reviewer 1 Report

Thank you for giving me this opportunity to review the manuscript. Please see attached.

Author Response

Dear editor Michael Tian,

We really appreciate all the comments of the reviewers. We answered point by point each comment and we highlighted in yellow the changes in the manuscript. Please see the attachment.

Reviewer 2 Report

This is an excellent paper from the technical point of view. Well written and explaining the significance of the study very clearly.

I am slightly hesitant to make some "typographical" suggestions or corrections. But here they are in the attached file. Many have to do with the use of the definite article (though US English usage may be claimed, that I find awkward). In some other cases there are spelling mistakes, all minor. In one or two cases I have suggested small adjustments to sentences to clarify the meaning. In the Abstract I think the statistics associated with the results distract from the meaning and recommend their removal.

Best wishes to the authors in their future collaborations.

Author Response

(The authors gave the same response as above.)

Reviewer 3 Report

The use of modern available communication technologies to influence the lifestyle of children is one of the effective ways to achieve this. The study under review is one of the first to evaluate longitudinally the network structure, the mechanisms of the formation and dissolution of friendships and the potential cohesion effect of a school-based governmental PA program in Latin America. The study shows that MARA, a school-based PA intervention that uses ICT, could promote social co-benefits that consist in increasing cohesion in the friendship network. From this point of view, the study is current and necessary. On the other hand, it is somewhat unbalanced. Much of the text is devoted to the description of models and their application. Unfortunately, significantly less is devoted to the discussion of the obtained results. In addition, the discussion is highly descriptive and does not analyze the data obtained. I also lack the limits of the study, both in terms of the application of statistical models and the explanation of the data obtained. Basic data were collected in 2013, this fact will need to be discussed in the discussion. The application of the models will be minimally affected by this, although at present the use of new technologies is significantly higher than seven years ago. Similarly, I recommend specifying the conclusion more and adding to the abstract the practical effects of the use of communication technologies in influencing children. The monitored group is morphologically and in terms of realized PAs considerably inhomogeneous, which should be discussed in the corresponding part of the study. Formally, BMI must have units. After reworking and supplementing the text, especially in the discussion chapter, I recommend publishing for publication.

Author Response

Dear editor Michael Tian,

We really appreciate all the comments of the reviewers. We answered point by point each comment and we highlighted in yellow the changes in the manuscript.
